# Moderating Role of Communication Competence in the Association between Professionalism and Job Satisfaction in Korean Millennial and Generation Z Nurses: A Cross-Sectional Study

**DOI:** 10.3390/healthcare11182547

**Published:** 2023-09-14

**Authors:** Young Jin Lee, Hyunjin Lee, Eun-Hi Choi

**Affiliations:** College of Nursing, Eulji University, Uijeongbu-si 11749, Republic of Korea; youngjin@eulji.ac.kr (Y.J.L.); hjlee@eulji.ac.kr (H.L.)

**Keywords:** MZ generation, communication competence, nursing professionalism, job satisfaction

## Abstract

Millennial and Generation Z (MZ generation) nurses, the core of the Republic of Korean nursing workforce, are leaving hospitals. We, therefore, aimed to determine the mediating role of communication competence between nursing professionalism and job satisfaction of MZ generation nurses in Republic of Korea. A total of 188 nurses aged 20–39 years belonging to the MZ generation, who had been working in a general hospital for over six months, participated in an online survey from 27 October 2022 to 11 January 2023. Data analysis was conducted using the bootstrapping method with the SPSS PROCESS macro program to confirm the mediating role of communication competence. Job satisfaction, professionalism, and communication skills were significantly positively correlated (r = 0.36–0.72, *p* < 0.001). Communication competence was found to be a mediating factor in the relationship between professionalism and job satisfaction. The results showed that when professionalism related to job satisfaction, good communication further enhanced job satisfaction. In order to enhance MZ generation nurses’ job satisfaction, it is necessary to improve nursing professionalism and implement training programs to improve communication skills considering generation-specific characteristics.

## 1. Introduction

The world is changing rapidly owing to the development of digital space, and millennials and Generation Z (hereinafter referred to as the ‘MZ generation’) are the most affected. The definition of the MZ generation varies among scholars; however, the millennial generation is generally considered to include individuals born between 1982 and 2000. Currently, they are aged 40 and below and make up a substantial portion of the healthcare workforce [1,2]. Generation Z is the direct successor to the millennial generation [3]. We used the term ‘MZ generation’ because millennials and Generation Z have commonalities in the way that they think and feel [4].

The MZ generation is sensitive to social changes and has the characteristics of embracing and respecting diversity while being individualistic and independent rather than blindly obeying authority [5]. This generation drew significant attention because of their behavior during the coronavirus disease 2019 (COVID-19) pandemic; in particular, these individuals tended to disregard social distancing requirements and local regulations [6]. In the learning field, the educational technology and feedback techniques used to focus the attention of MZ generation students need to differ from those used for previous generations [7]. In addition, in the workplace, job stability and job goals affect job retention for the previous generation, whereas low workload and satisfactory welfare affect job satisfaction for the MZ generation [8]. Owing to these differences from previous generations, the MZ generation is currently challenging leaders, managers, and educators in various fields [7].

As the baby boomer generation retires, the MZ generation will become the dominant workforce in the nursing profession [9]. In Republic of Korea, there is serious concern about the MZ generation nursing workforce, which has a tendency to leave healthcare institutions [8].

Aspects such as leadership, improvement of the work environment, professional growth, reduction of job-related fatigue, self-realization, and job satisfaction are relevant for preventing the attrition of nurses [10]. Job satisfaction is crucial because it serves as a motivating force for nurses to remain within an organization and enhances their nursing competencies [11]. In addition, improving nurses’ job satisfaction is key to improving work enthusiasm and reducing burnout and turnover intention [12,13]. During the COVID-19 pandemic, MZ generation nurses reported more burnout and lower job satisfaction compared to nurses from other generations [14,15,16]. Therefore, it is important to understand and evaluate this generation’s job satisfaction from a new perspective to prevent its nurses from leaving the workforce [10].

Factors influencing job satisfaction among nurses have been studied, and professional dedication has been found to be important in this regard [17]. Furthermore, enhancing professional values influences job satisfaction, leading to an impact on the intention to continue working [18]. In addition, a program that developed a professional identity was found to improve job satisfaction and nursing professionalism is needed to promote nurses’ internal motivation [19]. Particularly, nurses from the MZ generation perceive their profession as a factor in personal growth and development, and they emphasize the social value of their occupation. This social value encompasses the characteristic of being able to assist others and maintain amicable relationships with colleagues, as well as working in a vibrant work environment [20]. However, despite possessing such an optimistic occupational value perspective, it is noteworthy that their job satisfaction is lower compared to previous generations [21].

The job satisfaction of MZ generation nurses is influenced by relationships with colleagues, positive feedback from superiors, a favorable working environment, and communication [22]. While MZ generation nurses are independent, they value communication, seeking positive and close relationships [20,22]. They also have a low tolerance for incivility, and if they perceive a lack of respect or if negative relationships persist, they tend to leave their jobs easily [21]. 

During the COVID-19 pandemic, healthcare professionals experienced challenges in communication owing to infection control measures, such as mask-wearing and maintaining distance between medical staff. Consequently, effective communication is now strongly emphasized in the medical environment [23]. Previous studies have shown that interventions aimed at improving communication have led to enhancements in organizational teamwork and job satisfaction [24]. One reason for emphasizing communication is that preferred communication methods differ between generations. Baby boomers prefer face-to-face communication; Generation X uses email and face-to-face communication flexibly and Generation Y engages in continuous feedback through texts and messaging apps. The MZ generation prefers rapid and essential use of technology, making these individuals proficient in new media, primarily social media [10]. Differences in communication methods between generations can lead to errors in intergenerational communication [10,22]. In addition, nurses of the Baby boomer generation express their opinions more directly, while those of the MZ generation struggle to put up a polite appearance when communicating, and sometimes experience conflicts with nurses of other generations through unintentional expression of their feelings [22]. This may have an impact on the job satisfaction of MZ generation nurses as they work in clinical settings alongside diverse generations and provide nursing care to patients of different ages.

Against this background, we aimed to assess the nursing professionalism, communication competence, and job satisfaction levels of MZ generation nurses working in clinical settings. Additionally, we confirmed how the communication skills of MZ nurses, who use different communication methods from previous generations and react sensitively to others, affect their nursing professionalism and job satisfaction. By doing so, we intend to provide basic data for developing strategies to improve nurses’ job satisfaction. Therefore, the purpose of this study is to confirm the mediating role of communication ability in the relationship between professionalism and job satisfaction, and the specific aims are as follows:To assess the level of nursing professionalism, communication competence, and job satisfaction among MZ generation nurses.To examine differences in nursing professionalism, communication competence, and job satisfaction based on the characteristics of MZ generation nurses.To investigate the correlations between nursing professionalism, communication competence, and job satisfaction among MZ generation nurses.To confirm the mediating role of communication competence in the relationship between nursing professionalism and job satisfaction among MZ generation nurses.

## 2. Materials and Methods

### 2.1. Research Design

We used a descriptive cross-sectional design to examine the mediating role of communication competence in the relationship between nursing professionalism and job satisfaction.

### 2.2. Participants and Data Collection

The participants were nurses in their 20s and 30s (born after 1982) as of the data collection date. They had worked as nurses for more than six months in general hospitals with over 300 beds. A total of 188 individuals who understood the purpose of this study and voluntarily agreed to participate were included. The number of participants required was calculated using G*Power 3.1.9.7 [25]. For multiple regression analysis, with a significance level of 0.05, power of 0.95, medium effect size of 0.15, and 12 independent variables (10 participant characteristics and two independent variables), a minimum sample size of 184 was required. Considering a dropout rate of 20% owing to the online survey, 227 responses were collected. Of these, 39 incomplete responses were excluded, and the final analysis included 188 responses.

The data collection period was from 27 October 2022 to 11 January 2023. To collect data, flyers and URLs containing information about the study were posted in online communities and on social networking sites composed of nurses. Nurses accessed the URL, which included an explanation of the study and the online questionnaire. The explanation provided details about the research purpose, anonymity, and confidentiality of responses, the option to withdraw from the study at any time, and assurance that the collected data would be used solely for research purposes. Individuals who read the online explanation and clicked on the consent button were allowed to proceed with the online survey. The data collection process took approximately 15 min. In order to reduce the dropout rate due to incomplete responses, even if the online survey was interrupted, it could be continued by clicking the URL again later. In addition, as a token of appreciation, a beverage gift certificate worth $5 was provided to participants who completed the survey.

### 2.3. Research Tools

The research tools used in this study were approved by the original authors or the translators.

#### 2.3.1. Nursing Professionalism

The Nursing Professional Value Scale, developed by Yeun et al. [26], was used to measure nursing professionalism. This tool consists of 29 questions across five factors: self-concept of the profession, social awareness, professionalism of nursing, roles of nursing service, and originality of nursing. The instrument uses a 5-point Likert scale from 1 (Not at all) to 5 (Very much). Negative items are reverse scored, with higher scores indicating a more positive value of the nursing profession. Reliability, measured using Cronbach’s α, was 0.92 both in Yeun et al. [26] and this study.

#### 2.3.2. Communication Competence

The Global Interpersonal Communication Competence Scale, developed by Huh [27] to measure the communication competence of nurses and modified by Lee and Kim [28], was used. This instrument is composed of 15 subdimensions: self-disclosure, empathy and perspective-taking or dual perspective, social relaxation, assertiveness, concentration, interaction management, expressiveness, supportiveness, immediacy, efficiency, social appropriateness, conversational coherence, goal detection, responsiveness, and noise control. Items are rated on a 5-point Likert scale from 1 (Not at all) to 5 (Very much). Negative items are reverse scored, with higher scores indicating a higher level of communication ability. Cronbach’s α was 0.83 in Lee and Kim [28] and 0.86 in this study.

#### 2.3.3. Job Satisfaction

We used the Job Satisfaction Scale for Clinical Nurses developed by Lee et al. [29]. This tool consists of a total of 33 items grouped into six areas: (1) recognition from the organization and professional achievement; (2) personal maturation through the nursing profession; (3) interpersonal interaction with respect and recognition; (4) accomplishment of accountability as a nurse; (5) display of professional competency and (6) stability and job worth. The tool is rated on a 5-point Likert scale from 1 (Not at all) to 5 (Very much), with higher scores indicating higher levels of job satisfaction. Cronbach’s α was 0.95 in both Lee et al. [29] and this study.

#### 2.3.4. General Characteristics

The assessed general characteristics included gender, age, marital status, educational level, salary, work experience, work type, job position, experience of job turnover, and working department.

### 2.4. Data Analysis

The data were analyzed using SPSS 24.0 (IBM Corp., Armonk, NY, USA). Participant characteristics, nursing professionalism, communication competence, and job satisfaction were analyzed using descriptive statistics. Differences between variables based on participants’ general characteristics were analyzed using independent *t*-tests and one-way analysis of variance. Post hoc testing was conducted using Scheffe’s test. Correlations between variables were analyzed using Pearson’s correlation coefficient. To test the mediating role of communication competence in the relationship between professional intuition and job satisfaction, we used the SPSS PROCESS macro model 14 [30]. To verify the mediating effect, bootstrapping was used, presenting significance testing through confidence intervals. Bootstrapping involves analyzing by resampling k times from the sample data, without assuming the distribution shape of the mediating effect. We re-extracted 5000 bootstrap samples to analyze the indirect effect with a 95% confidence interval.

### 2.5. Ethical Considerations

Prior to data collection, approval was obtained from the Institutional Bioethics Committee of Eulji University (IRB No: EU22-58). All questionnaires were administered after obtaining voluntary consent, and anonymity was ensured. The participants were given a detailed explanation of the study, voluntarily participated, and could stop answering the questionnaire at any point. The researcher’s contact information was provided so that those who had inquiries about the study could reach out at any time during data collection.

## 3. Results

### 3.1. Participant Characteristics

Of the participants 63.8% were women, 55.3% were in their 30s, and the mean age was 30.2 years. Further, 70.7% were unmarried, 69.1% had a bachelor’s degree, and 59.6% earned less than KRW 50 million (approximately 38,000 USD) per year. The mean working period was five years, and 54.3% of the participants had worked for two to five years. Of the participants, 74.5% were working shifts, 92.6% were staff nurses, 39.4% were working in a ward, and 30.3% had experienced a job change (Table 1).

### 3.2. Level of Nursing Professionalism, Communication Competence, and Job Satisfaction

The mean nursing professionalism score was 3.30 out of 5. The professionalism of nursing category scored the highest (mean: 3.63 points), and the social awareness category scored the lowest (mean: 2.84 points). The mean communication competence score was 3.73 out of 5, and the mean job satisfaction score was 3.44 out of 5. The accomplishment of accountability as a nurse category scored the highest (mean: 3.94 points), and recognition from the organization and professional achievement category scored the lowest (mean: 3.10 points) (Table 2).

### 3.3. Nursing Professionalism, Communication Competence, and Job Satisfaction According to Participant Characteristics

There was a significant difference in nursing professionalism based on participants’ work department (F = 3.78, *p* = 0.011). Nurses working in the operating room and the laboratory had the highest scores, while those working in the intensive care unit had the lowest scores. Communication competence differed significantly based on gender (t = 2.02, *p* = 0.046). Male nurses had significantly higher communication ability scores than female nurses. Job satisfaction differed significantly based on several factors. Married nurses (t = 3.33, *p* = 0.001), nurses receiving a salary of 50 million KRW or more (t = 2.74, *p* = 0.007), and nurses who worked regular hours reported significantly higher job satisfaction (t = 2.39, *p* = 0.018). Additionally, job satisfaction varied according to the work department, with nurses in the operating room and laboratory reporting the highest levels, and nurses in the intensive care unit reporting the lowest levels (F = 3.20, *p* = 0.024) (Table 3).

### 3.4. Correlation between Nursing Professionalism, Communication Competence, and Job Satisfaction

Nursing professionalism showed significant positive correlations with communication competence (r = 0.36, *p* < 0.001) and job satisfaction (r = 0.72, *p* < 0.001), that is, higher levels of nursing professionalism were associated with higher levels of communication competence and job satisfaction. Additionally, there was a positive correlation between communication competence and job satisfaction (r = 0.56, *p* < 0.001) (Table 4).

### 3.5. Mediating Role of Communication Competence between Nursing Professionalism and Job Satisfaction

Using the bootstrapping method, we assessed the mediating role of communication competence between nursing professionalism and job satisfaction. General characteristics that exhibited significant differences based on each variable, such as gender, marital status, salary, work type, and working department, were treated as control variables. From the mediation analysis, the direct effect of nursing professionalism on job satisfaction was 0.67. The indirect effect attributed to communication competence was 0.15. Both the direct and indirect effects were considered significant as they did not encompass 0 within the 95% confidence interval, from the lower limit to the upper limit. This indicates that as communication competence rose, it significantly and positively enhanced job satisfaction, which was statistically significant (Table 5). 

## 4. Discussion

In this study, we aimed to explore the mediating role of communication competence in the relationship between nursing professionalism and job satisfaction among MZ generation nurses, who account for most of the nursing workforce in Republic of Korea. We identified the factors associated with job satisfaction among MZ generation nurses, with a focus on their communication competence, and confirmed that communication competence played a mediating role in the relationship between nursing professionalism and job satisfaction. These findings highlight the importance of incorporating communication training into future programs designed to enhance job satisfaction among nurses and provide meaningful insights for nursing managers.

The level of job satisfaction of MZ generation nurses in this study was 3.44 points, similar to the score of 3.32 points measured using the same tool among nurses aged 20 to 39 years working in general hospitals, but lower than 3.71 points for nurses aged 40 years and older [31]. Previous studies have found that job satisfaction increases significantly with age [31,32,33]. As job satisfaction differs across generations, it is important to understand the characteristics of job satisfaction in MZ generation nurses. It is important to increase the job satisfaction of MZ generation nurses with little job experience, who make up most of the hospital medical staff. The improvement in job satisfaction of nurses of this generation is likely to have a positive effect on their intention to stay in the job.

In this study, the mean sub-domain scores in areas related to occupational expertise such as ‘accomplishment of accountability as a nurse’ and ‘stability and job worth’ were high, whereas the ‘recognition from the organization and professional achievement’ and ‘interpersonal interaction with respect and recognition’ sub-domain scores were low. These findings suggest that participants believed that they were not sufficiently compensated for their work and that their abilities were not adequately recognized by members within the organization. According to a policy report on the characteristics of the MZ generation in Republic of Korea, this generation wants immediate feedback and has a strong desire for recognition of their work; therefore, if their performance review is perceived as unfair, their enthusiasm and motivation for work decreases [34]. These characteristics are also evident in MZ generation nurses in Western countries [32,33,35]. Therefore, it is important for older nurses to respect the contributions of younger nurses to their organizations and fully acknowledge their abilities. The workplace should foster an environment that rewards employees appropriately. In this study, higher salary and working fixed hours were associated with higher job satisfaction and working in the intensive care unit was associated with lower job satisfaction, supporting the results of previous studies. The MZ generation highly values personal growth and development and places importance on recognition and mutual respect from organizations [36,37]. Mutual respect is fostered through open and trusting communication [38]. Nurse managers should, therefore, create a work environment that is open, honest, and respectful and provide opportunities for active participation in communication for MZ generation nurses.

The communication competency score was 3.73, indicating a medium to high level. This was higher than the 3.44 points [39] scored by general hospital nurses in previous studies using the same tool but lower than the 3.92 points scored by nurses with an average of 23 years of work experience [40]. The participants in this study were aged 20–39 years, with an average work experience of approximately five years. It is likely that the MZ generation finds it difficult to communicate with patients or medical staff face-to-face because they are accustomed to using mobile devices and social networking sites. Previous studies have found differences in communication ability according to work duration [39]. However, as we only surveyed the MZ generation, there are some limitations in confirming differences in communication competencies according to age. Therefore, further studies should include nurses of all ages to identify communication characteristics and their variations across generations. 

We also found that communication competency differed according to gender and that male nurses’ communication competency scores were higher than those of female nurses. This is consistent with the finding that male students have stronger communication skills than female students reported in a study of nursing students who used the same tool [41]. However, given the smaller number of male nurses in this study compared to female nurses, there is a need for further research including more male participants. Additionally, different assessment tools measuring gender differences in communication abilities have shown varying results [42,43], indicating that a simple comparison of communication skills between male and female nurses might be limited. Communication is a two-way process and receiving positive feedback can lead to changes in perception and improvement in communication competence. Therefore, to enhance communication competence, it is necessary to develop diverse educational programs tailored to the participants’ gender, age, and other individual sociodemographic characteristics.

Upon analyzing the scores on items of communication competence, it was found that the participants scored high on ‘social appropriateness’ (using polite language and choosing words suitable for the interlocutor and situation) and ‘responsiveness’ (reacting sensitively to the other party’s reactions, such as making eye contact and nodding). On the contrary, they scored low on ‘conversational coherence’ and ‘social relaxation’, which pertains to behaving calmly and composedly in interpersonal relationships without feeling anxiety or fear and being able to handle negative reactions or criticisms from others appropriately. This can be seen as reflecting the characteristics of the MZ generation, who are sensitive to rudeness from others tend to avoid conversations, and show negative reactions when they cannot respond appropriately [20,21]. Therefore, to enhance the communication competence of MZ generation nurses, there is a need for training in mind control to remain unfazed in difficult situations and in communication skills to handle such situations appropriately. In particular, there is a need to provide assertiveness training that enables them to express their emotions and opinions persuasively.

The score for nursing professionalism was 3.30 points out of 5, similar to the score of 3.39 points [44] in a previous study that used the same tool. Participants in this study recognized the professionalism of nursing as the highest and social awareness as the lowest among the subfactors of nursing professionalism. Social awareness is reported to be the lowest of the aspects of nursing professionalism in previous studies in Republic of Korea [44,45], which can be interpreted as indicating that Republic of Korean nurses have low social awareness of nursing. If nurses are portrayed with a contradictory image of society, their beliefs, values, and self-esteem can be damaged. Therefore, to enhance nursing professionalism among nurses, there is a need to positively change the social perception of nursing and nurses. This could help foster a more positive and supportive environment for nurses and promote their professional values and self-worth. Despite the recent COVID-19 pandemic, which has strengthened society’s positive perception of the necessity and expertise of nurses, it appears that nurses working in healthcare settings still perceive low societal recognition. Therefore, support is needed not only from within the hospital, involving other healthcare professionals, but also from the external socio-political aspects of society to improve the recognition and remuneration for nurses’ professionalism.

We found that communication competency mediated the role of nursing professionalism with regard to job satisfaction. In other words, the higher the nursing professionalism, the higher the communication competency and, ultimately, job satisfaction; the explanatory power of these variables with regard to job satisfaction was 82%. It was confirmed that nurses’ communication competency is a mediating variable between nursing professionalism and job satisfaction and is necessary for nurses to convert nursing professionalism to job satisfaction. Nursing professionalism involves perceiving and understanding human beings, making judgments about real-life nursing situations, and providing value to nursing actions [46]. This is closely related to the years of clinical experience and education [47]. The positive perception of such nursing expertise by nurses is crucial because it positively influences communication with patients and colleagues [44]. Nurses’ communication competency is an essential factor for identifying patients’ needs and solving their problems and is necessary for all interpersonal relationships in the medical field, including relationships with other medical staff and family members, as well as relationships with patients [48]. As nurses improve their communication competency, their emotional stress decreases [39]. Additionally, successful problem-solving and coping strategies can enhance clinical performance [40]. Furthermore, nurses can improve patient satisfaction through effective communication [49], which in turn can lead to increased job satisfaction [31,39]. Therefore, nursing managers should prioritize educating MZ generation nurses about the importance of improving communication skills and emphasize the cultivation of nursing professionalism and communication competence. Clinical settings include patients and healthcare workers of various ages, including baby boomers, and Generations X, M, and Z [31,32,33]. Therefore, nursing managers should explore the differences in communication characteristics by generation. As intergenerational conflict can increase when communication styles differ [50], efforts should be made to minimize misunderstandings related to intergenerational differences. Communication programs for nurses in the clinical field should be actively developed and implemented. Previous studies have found that communication competency is improved using standardized scenarios and simulations focusing on specific patient situations [51,52,53] and that participating in such a communication program increases nurses’ job satisfaction as well as job competency [51].

Some variables with significant differences in nursing professionalism, communication ability, and job satisfaction by general characteristics were not related to job satisfaction in the mediation analysis. However, these variables may have important implications and require further investigation. For example, gender, marital status, salary, work type, and working department were not related to job satisfaction. Although our data collection period was during the COVID-19 pandemic, the working department was not related to job satisfaction. Previous studies have reported that increasing awareness of the threat of COVID-19 or working in a high-risk environment such as an intensive care unit could affect nurses’ emotional fatigue [14]. Our results also showed that nurses in the operating room or laboratory departments had higher job satisfaction than those in the intensive care unit. Therefore, it is necessary to conduct repeated studies with nurses of different levels in different regions to confirm the factors influencing these variables.

During the COVID-19 pandemic, young nurses were found to be vulnerable to burnout [14], which is reported to have a direct impact on job satisfaction. However, the job satisfaction of MZ generation nurses showed a similar level to the job satisfaction measured using the same tool before COVID-19 [29,31]. As we targeted only the MZ generation and there are limitations in comparatively analyzing job satisfaction by age directly owing to COVID-19, it seems necessary to evaluate job satisfaction through repeated studies.

This study has some limitations. We collected data from nurses working in various hospitals using an online questionnaire. As a result, there may have been differences in hospital size, employee benefits, working environment, and staffing structure among the participating nurses. Owing to these potential differences, caution is necessary when interpreting the results. Additionally, we specifically targeted the MZ generation, limiting the ability to compare nursing professionalism, communication competence, and job satisfaction across generations. Therefore, it would be useful to conduct further research that includes nurses of diverse generations to investigate and compare these characteristics across different generations. Finally, while the young MZ generation was expected to experience a high level of burnout during COVID-19, we did not factor this into the analysis. Burnout due to COVID-19 can affect communication and job satisfaction. Therefore, there are limitations in interpreting causal relationships in our study. Accordingly, some design changes are warranted for future research on professionalism, communication ability, and job satisfaction of MZ generation nurses. Research can be conducted including the influencing factors of various variables including nurse burnout. This will contribute to the investigation of more complex factors.

## 5. Conclusions

We attempted to identify ways to improve communication competency and job satisfaction among MZ generation nurses in Republic of Korea by confirming the mediating role of communication competency between nursing professionalism and job satisfaction. The results confirmed that communication competency mediates the relationship between nursing professionalism and job satisfaction. Therefore, to understand the MZ generation and reflect the characteristics of MZ nurses, who value relationships and avoid conflict situations, it is deemed necessary to develop a specialized program to enhance their distinct communication abilities. Training is required to effectively express one’s opinions and to appropriately handle various situations, including potential negative situations that can arise during interactions between staff, patients, and guardians, without feeling tense. Moreover, as differences in communication based on gender were observed, it is recommended to conduct follow-up studies to understand the detailed characteristics and types of communication according to gender and to develop programs considering gender. Based on the results of this study, we recommend the following. First, additional research is needed to compare the differences in nursing professionalism, communication competence, and job satisfaction among nurses of different generations and medical facility sizes. Second, it is necessary to develop a program to improve nursing professionalism suitable for MZ generation nurses and to verify its effect. Third, research is needed to develop an education program to improve communication competency among MZ generation nurses and verify its effectiveness. Overall, by addressing these recommendations, it should be possible to create a more supportive and fulfilling work environment for MZ generation nurses and improve their overall job satisfaction.

## Figures and Tables

**Table 1 healthcare-11-02547-t001:** Characteristics of participants (N = 188).

Variable	Categories	n (%)	Mean ± SD	Range
Gender	Men	68 (36.2)		
	Women	120 (63.8)		
Age (years)			30.2 ± 3.6	23–39
	≤29	84 (44.7)		
	30–39	104 (55.3)		
Marital status	Unmarried	133 (70.7)		
	Married	55 (29.3)		
Educational level	Diploma	21 (11.2)		
	Bachelor	130 (69.1)		
	Graduate	37 (19.7)		
Salary (KRW)	<50 million ^1^	112 (59.6)		
	≥50 million	76 (40.4)		
Work experience (years)			5.0 ± 3.5	1–20
	≤1	20 (10.6)		
	2–5	102 (54.3)		
	6–10	54 (28.7)		
	≥11	12 (6.4)		
Work type	Shift work	140 (74.5)		
	Regular working hours	48 (25.5)		
Job position	Staff nurse	174 (92.6)		
	Charge nurse	12 (6.4)		
	Head nurse	2 (1.1)		
Working department	Ward	74 (39.4)		
	ICU	38 (20.2)		
	OR & Laboratory	33 (17.6)		
	ED & OPD	43 (22.9)		
Experience of job turnover	No	131 (69.7)		
	Yes	57 (30.3)		

^1^ 50 million KRW is equivalent to approximately 38,000 USD. ED, emergency department; ICU, intensive care unit; OPD, outpatient department; OR, operating room.

**Table 2 healthcare-11-02547-t002:** Level of nursing professionalism, communication competence, and job satisfaction (N = 188).

Variable	Possible Range	Mean ± SD	Actual Range
**Nursing professionalism**	1–5	3.30 ± 0.52	2.00–4.83
Self-concept of the profession		3.48 ± 0.55	1.67–5.00
Social awareness		2.84 ± 0.71	1.25–4.50
Professionalism of nursing		3.63 ± 0.59	2.00–5.00
Roles of nursing service		3.60 ± 0.61	2.00–5.00
Originality of nursing		3.07 ± 0.71	1.00–5.00
**Communication competence**	1–5	3.73 ± 0.46	2.67–5.00
Self-disclosure		3.58 ± 0.82	1–5
Empathy and perspective-taking or dual perspective		3.97 ± 0.73	1–5
Social relaxation		3.32 ± 0.86	1–5
Assertiveness		3.55 ± 0.87	2–5
Concentration		3.94 ± 0.70	1–5
Interaction management		3.42 ± 0.86	1–5
Expressiveness		3.52 ± 0.80	1–5
Supportiveness		3.63 ± 0.91	1–5
Immediacy		3.80 ± 0.68	2–5
Efficiency		3.71 ± 0.84	2–5
Social appropriateness		4.28 ± 0.73	2–5
Conversational coherence		3.39 ± 0.83	2–5
Goal detection		3.89 ± 0.68	2–5
Responsiveness		4.13 ± 0.67	2–5
Noise control		3.85 ± 0.75	2–5
**Job satisfaction**	1–5	3.44 ± 0.59	1.94–5.00
Recognition from the organisation and professional achievement		3.10 ± 0.76	1.11–5.00
Personal maturation through the nursing profession		3.34 ± 0.82	1.00–5.00
Interpersonal interaction with respect and recognition		3.42 ± 0.59	2.00–5.00
Accomplishment of accountability as a nurse		3.94 ± 0.60	1.75–5.00
Display of professional competency		3.69 ± 0.64	2.33–5.00
Stability and job worth		3.83 ± 0.69	2.00–5.00

**Table 3 healthcare-11-02547-t003:** Nursing professionalism, communication competence, and job satisfaction according to participant characteristics (N = 188).

		Nursing Professionalism		Communication Competence		Job Satisfaction	
Variables	Categories	Mean ± SD	t/F (*p*)	Mean ± SD	t/F (*p*)	Mean ± SD	t/F (*p*)
Gender	Men	3.31 ± 0.49	0.52	3.83 ± 0.52	2.02	3.45 ± 0.57	0.29
	Women	3.27 ± 0.56	(0.602)	3.68 ± 0.41	(0.046)	3.43 ± 0.62	(0.776)
Age (years)	≤29	3.30 ± 0.52	1.11	3.77 ± 0.48	1.15	3.39 ± 0.54	1.17
	30–39	3.30 ± 0.51	(0.912)	3.70 ± 0.44	(0.251)	3.49 ± 0.62	(0.246)
Marital status	Unmarried	3.26 ± 0.50	1.46	3.72 ± 0.47	0.47	3.35 ± 0.59	3.33
	Married	3.39 ± 0.54	(0.146)	3.76 ± 0.43	(0.639)	3.66 ± 0.54	(0.001)
Educational level	Diploma	3.43 ± 0.45	1.49	3.74 ± 0.40	0.12	3.47 ± 0.57	0.04
	Bachelor	3.31 ± 0.52	(0.229)	3.74 ± 0.49	(0.889)	3.44 ± 0.59	(0.966)
	Graduate	3.19 ± 0.52		3.70 ± 0.39	1.04	3.45 ± 0.60	
Salary (Won × 10^4^)	<5000	3.29 ± 0.50	0.24	3.71 ± 0.49	0.97	3.35 ± 0.57	2.74
	≥5000	3.31 ± 0.54	(0.814)	3.77 ± 0.40	(0.335)	3.58 ± 0.60	(0.007)
Work experience (years)	≤1	3.30 ± 0.55	0.83	3.70 ± 0.38	0.72	3.40 ± 0.65	2.19
	2–5	3.26 ± 0.54	(0.477)	3.78 ± 0.48	(0.540)	3.37 ± 0.59	(0.091)
	6–10	3.35 ± 0.40		3.67 ± 0.40		3.54 ± 0.52	
	≥11	3.47 ± 0.69		3.71 ± 0.56		3.76 ± 0.65	
Work type	Shift work	3.26 ± 0.51	1.64	3.72 ± 0.46	0.56	3.38 ± 0.57	2.39
	Regular working hours	3.40 ± 0.54	(0.104)	3.76 ± 0.44	(0.576)	3.62 ± 0.60	(0.018)
Job position	Staff nurse	3.31 ± 0.51	0.64	3.73 ± 0.46	0.34	3.44 ± 0.58	0.45
	Charge nurse	3.20 ± 0.58	(0.529)	3.79 ± 0.42	(0.693)	3.58 ± 0.77	(0.642)
	Head nurse	2.98 ± 0.51		3.50 ± 0.24		3.23 ± 0.49	
Working department	Ward (a)	3.22 ± 0.52	3.78	3.76 ± 0.51	0.81	3.45 ± 0.56	3.20
	ICU (b)	3.16 ± 0.50	(0.011)	3.63 ± 0.43	(0.492)	3.21 ± 0.60	(0.024)
	OR & Laboratory (c)	3.50 ± 0.49	c > b *	3.77 ± 0.49		3.60 ± 0.59	c > b *
	ED & OPD (d)	3.40 ± 0.49		3.74 ± 0.34		3.52 ± 0.59	
Experience of turnover	No	3.33 ± 0.48	1.25	3.73 ± 0.43	0.64	3.47 ± 0.58	1.04
	Yes	3.23 ± 0.60	(0.212)	3.73 ± 0.52	(0.949)	3.38 ± 0.60	(0.299)

* Scheffe’s test.

**Table 4 healthcare-11-02547-t004:** Correlation between nursing professionalism, communication competence, and job satisfaction (N = 188).

Variables	Nursing Professionalism r (*p*)	Communication Competence r (*p*)	Job Satisfaction r (*p*)
Nursing professionalism	1.00		
Communication competence	0.36 (<0.001)	1.00	
Job satisfaction	0.72 (<0.001)	0.56 (<0.001)	1.00

**Table 5 healthcare-11-02547-t005:** Mediating role of communication competence between nursing professionalism and job satisfaction (N = 188).

Variables		Effect	SE	95% CI	
				LLCI	ULCI
Direct	NP → JS	0.67	0.05	0.56	0.78
Indirect	NP → CC → JS	0.15	0.03	0.08	0.21
Total		0.82	0.06	0.70	0.93
	R^2^	0.58			
	F	42.12			
	*p*	<0.001			

NP = nursing professionalism; JS = job satisfaction; CC = communication competence; CI = confidence interval; LLCI = lower limit of B in 95% confidence interval; ULCI = upper limit of B in 95% confidence interval.

## Data Availability

The datasets generated for this study are available upon request from the corresponding author.

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
