# Peer review of "Moderating Role of Communication Competence in the Association between Professionalism and Job Satisfaction in Korean Millennial and Generation Z Nurses: A Cross-Sectional Study"

_healthcare, 2023, doi:10.3390/healthcare11182547_

Round 1

Reviewer 1 Report

This study has a merit due to the study population, MZ Generation nurses. There are some suggestions for improvement:

1. In introduction, current problems why MZ Generation nurses need to be studied are not been displayed. There are already many studies that evaluated the nursing professionalism, communication competence, and job satisfaction. Why did the authors studied this concepts among MZ Generation. What were their problem? And why did the authors selected Baron and Kenny's three-step mediating effect to investigate these concepts?

2. Study target was clinical nurses aged 20 to 39 years. The range of the age are 20 years. 

3. One of the  criticisms about Baron and Kenny method is that the mediating effect differs by size of the study population.  The study population of this study were 188(nearly 20% drop out). Please describe what kind of effect did the author put to prevent the dropout.

4. Unreliable and rigid nature of the Sobel Test are among the most substantive criticisms about Baron and Kenny approach.  Currently (from the year 2010), bootstrapping method the increasingly well-accepted mediation analysis approach (Bollen and Stine, 1990; Fritz and MacKinnon, 2007; Hayes, 2018; Hayes and Rockwood, 2017; MacKinnon, Fairchild and Fritz, 2007; MacKinnon et al., 2004; Shrout and Bolger, 2002; Preacher and Hayes, 2004, 2008b; Preacher and Selig, 2012; Williams and MacKinnon, 2008).

5. In table 3, it showed that there were a significant differences in nursing professionalism, communication competence, and job satisfaction by general characteristics. Therefore, without adjusting these variables, it is hard to insist that the communication competence played the partial mediating effect.

There are some English grammar errors.  An edit by a native English speaker is needed

Author Response

Thank you for your helpful comments on this manuscript. We tried to address the issues that you pointed out in your comments as much as possible, and the revised parts of the manuscript are shown in red font in the Word file. In addition, point-by-point responses to your comments are as follows:

Reviewer 2 Report

.Dear authors, 

first of all thank you for this amazing and innovative study. Your results show important contributes to nursing education, but also to manage nursing services. 

I have just two suggestions to improve your manuscript: 

1) Can you identify in the conclusions areas of communication skills that should be developed in nursing training programs, regarding the results of your study? 

2) Considering the differences identified between male and female nurses regarding communication skills, ,can you give any more information in the conclusions about this issue? 

Congrats again, it was an honour to review your manuscript. 

Good work! 

You have some litle mistakes that I suggest to review! Nothing special but as you are going to improve the manuscript, make it perfect. For e.g.: In line 188, where it is written: ", a higher the salary" should be " , a higher salary"

Author Response

(The authors gave the same response as above.)

Reviewer 3 Report

Dear Authors,

It's a useful and interesting work on the interaction between communication competency and nursing professionalism, and job satisfaction in nurses.

The study is well-designed, and the results are interesting. In my opinion, there are only some minor points to be enlightened:

In the discussion section:

-       Among the results, it stands out that the higher the age, the higher the professional satisfaction, as in previous studies, but the influence of the covid-19 pandemic on the variables studied has yet to be discussed in this study. The population studied is young, and there are studies that say that sociodemographic factors such as younger age are a risk factor that increased nurse burnout during the COVID-19 pandemic (Galanis et al., 2021. doi: 10.1111/jan.14839). This could directly influence professional satisfaction.

-       On the other hand, it would have been interesting to measure nurse burnout since its presence could be influencing nurses' personal fulfillment and communication with patients in a situation in which the Covid pandemic made depersonalization greater. Not having measured it constitutes a limitation of the study that should be reflected.

-       It is also striking that the results of this study show different levels of job satisfaction according to the work department without having pointed out the possible influence of the pandemic. This should be discussed. There are studies that say that a greater perception of COVID-19 threats or working in a high-risk environment such as the ICU could influence nurses' emotional exhaustion, and this could justify that in this study, nurses in the operating room or laboratory departments reported higher levels of job satisfaction than those in the intensive care unit.

Author Response

(The authors gave the same response as above.)

Round 2

Reviewer 1 Report

The manuscript has met the quality of this journal. Congratulation for your work.